# Sesquiterpene Induction by the Balsam Woolly Adelgid (*Adelges piceae*) in Putatively Resistant Fraser Fir (*Abies fraseri*)

Austin Thomas [1,*], David C. Tilotta [2], John Frampton [3] and Robert M. Jetton [3]

[1] Department of Forestry and Environmental Resources, USDA Forest Service De Soto Forest, University of Kentucky, Formerly North Carolina State University, Asheville, NC 28804, USA

[2] Department of Forest Biomaterials, North Carolina State University, Saucier, MS 39574, USA; dctilott@ncsu.edu

[3] Department of Forestry and Environmental Resources, North Carolina State University, Raleigh, NC 27607, USA; frampton@ncsu.edu (J.F.); rmjetton@ncsu.edu (R.M.J.)

[*] Correspondence: austin.thomas@uky.edu

**Abstract:** Fraser fir, *Abies fraseri* (Pursh) Poir., is a tree endemic to the Southern Appalachians and is found only in a few isolated populations at high elevations. Fraser firs are also cultivated on a commercial scale as Christmas trees. The species is imperiled by an introduced insect, the balsam woolly adelgid, *Adelges piceae* Ratzeburg (BWA). The insect severely damages Christmas tree crops and has caused substantial Fraser fir mortality in natural stands. Foliar terpenoids are one mechanism of host plant defense against invading insects and may be one focus of future Christmas tree breeding efforts. This study examines the correlation of foliar terpenoids with Fraser fir performance when infested with BWA. GC-MS and GC-FID analysis of artificially infested Fraser fir foliage reveals that increased concentrations of four terpenoid compounds are associated with BWA infestations. Foliar concentrations of two sesquiterpenes, camphene and humulene, are significantly higher in putatively resistant Fraser fir clones than in more susceptible clones after sustained adelgid feeding for a period of 20 weeks. Although it is unclear if the induction of these sesquiterpenes in the host fir is directly contributing to adelgid resistance, these compounds could serve as effective indicators while screening for BWA resistance in future Christmas tree breeding programs.

**Keywords:** Fraser fir; *Abies fraseri*; balsam woolly adelgid; *Adelges piceae*; terpenoids; resistance

## 1. Introduction

Plants are sessile primary producers and are therefore immensely vulnerable to biotic attack. In response to herbivore attacks, plants have evolved complex defense responses in the form of structural barriers, the production of toxic chemicals, and the attraction of natural enemies [1]. Volatile chemical emissions known as herbivore-induced plant volatiles (HIPVs) are the primary mechanism through which plants signal for natural enemies. Some HIPVs may also be directly toxic to herbivorous insects [2]. Insect response to vegetation odors and phytochemistry varies significantly from species to species [3]. HIPV profiles, likewise, vary according to the plant, the plant's phenology, and to the attacking herbivore species [4].

The evolution of plant defensive responses is often mediated by a few keystone herbivores which are abundant in the host plant's environment [5]. These coevolved defensive responses are more effective at defending the host plant from native herbivores than from introduced herbivores, hence the often detrimental outcome of introduced insects feeding on a novel host [6]. For example, eastern hemlock, *Tsuga canadensis* (L.) Carriére, is currently threatened by the invasive, xylem-feeding, hemlock woolly adelgid, *Adelges tsugae* Annand (HWA). The chemical defense mounted by the majority of eastern hemlocks in response to HWA feeding is not effective. However, this same defensive response is effective against native defoliating insect species [7,8]. Variation in volatile

defenses within species still exists in nature despite the evolutionary pressure exerted by keystone herbivores and this variation often has a genetic basis [9]. Plant breeders can take advantage of this natural variation within a host plant species to select for individuals with more appropriate defensive responses to novel insect herbivores [10].

Fraser fir, *Abies fraseri*, (Pursh) Poir., is a conifer endemic to the Southern Appalachians where it is only found in natural stands at high elevations in mixed spruce-fir forests [11]. Fraser fir is also a specialty crop grown on a large scale as Christmas trees at middle elevations (600–1200 m) in the region. The Fraser fir is particularly susceptible to attack by the introduced balsam woolly adelgid, *Adelges piceae*, Ratz. (BWA), and experiences a range of adverse responses to feeding by the insect [12]. Symptoms of BWA feeding on Fraser fir include a swelling of the terminal branches called gout, a loss of apical dominance, reduced growth rates, and the systemic formation of abnormal wood called rotholz [13,14]. These symptoms weaken the host fir and contribute to significant mortality in natural Fraser fir stands, likely in conjunction with environmental stressors such as drought [13]. Beyond BWA's association with increased fir mortality in natural stands, gouting and loss of apical dominance caused by BWA infestation greatly reduce the market value of Christmas trees.

No major predators or parasitoids of BWA are known from the Southern Appalachians, and all attempts at classical biological control (importation and release of natural enemies from the native range of an invasive species) to curb BWA outbreaks in the region have thus far been ineffective [15,16]. Ongoing BWA infestations in natural stands of Fraser fir are projected to reduce the genetic diversity of the species, imperiling its long-term survival [16]. Efforts to find and propagate adelgid-resistant Fraser firs and to conserve the genetic diversity of the tree are needed to restore and maintain Southern Appalachian spruce-fir forests. These measures could also reduce the time and financial costs to the Christmas tree industry currently required to control adelgids in Fraser fir plantations.

Terpenes and their derived terpenoids are among the most prevalent HIPVs in the plant kingdom. These compounds have diverse roles in plant defense against insects: acting as toxicants, repellents, elicitors of other defense related pathways, and as mediators of indirect defenses via predators or parasitoids [17]. Previous work by Carlow et al. [18], which compared the volatile foliar chemistry of infested and uninfested wild Fraser firs, found an increase in the monoterpenes 3-carene, β-pinene, and the sesquiterpenes β-caryophyllene, humulene, and β-bisabolene in select infested trees. This subset of terpenes may provide some resistance against the balsam woolly adelgid [18]. In a different study, Grégoire et al. [19] found an inverse relationship between BWA-caused gouting and the monoterpenes α-pinene and camphene in the closely related Balsam fir, *Abies balsamea*, (L.) Mill. Finally, the sesquiterpenoid compound juvabione, derived from α-bisabolene, is present in varying quantities in Fraser fir bark. Juvabione is a juvenile-hormone-related compound and its concentrations are positively correlated with the retention of apical dominance in the host and negatively correlated with BWA fecundity [20].

In this study, we chromatographically and statistically characterize a subset of fir HIPVs primarily consisting of extractible terpenoids. Furthermore, we relate the foliar concentration of these volatiles to Fraser fir's susceptibility to BWA. This work aids in an ongoing effort to integrate BWA resistance breeding into existing Fraser fir Christmas tree breeding programs in the state of North Carolina. Additionally, knowledge gained through this research may prove useful in the effort to protect wild Fraser fir populations from new waves of BWA-caused mortality by identifying HIPV profiles associated with host resistance. Thus, genetic conservation efforts can be directed towards trees with some level of host resistance to BWA attack.

## 2. Materials and Methods

A recent study by Thomas [21] using grafted Fraser fir clones in pots sourced from the Great Smokey Mountains National Park and from first generation selections, made by the North Carolina Premium Fraser Fir Seed Cooperative Orchard, reported significant

differences in the levels of BWA resistance across 37 clones. Clones sourced from the seed cooperative orchard for this study were chosen at random and had never undergone selection or testing for BWA resistance prior to their inclusion in this study. The clones were artificially infested with BWA in 2016 and underwent a performance assessment in 2018. A subset of 14 clones from the initial study were reinfested in 2019 for further evaluation. HIPV analysis was also completed on these 14 clones in 2019. The results of the HIPV analysis are reported here as a separate communication because it builds upon previous literature suggesting specific mono and sesquiterpenes may be associated with some degree of BWA resistance in Fraser fir.

### 2.1. Foliage Collection

The study area, experimental design, artificial infestation of Fraser firs with BWA, and Fraser fir clone performance assessments are described in full detail in Thomas, 2021 [21], and briefly summarized here. The study was conducted at the North Carolina Department of Agriculture and Consumer Services Mountain Research Station in Waynesville, NC, USA (35°29′15.072″ N, 82°58′03.648″ W elevation 820 m) in 2019. The study consisted of 12 clones of Fraser fir from the North Carolina Premium Fraser Fir Seed Cooperative Orchard and the Great Smokey Mountains National Park as well as one clone each of unimproved Silver fir and Veitch fir. Silver fir is partially tolerant to BWA, while Veitch fir has a very high degree of resistance to the insect [22,23]. A total of 92 grafted ramets of Fraser fir were artificially infested on 17 May 2019 via the suspended bolt method [14], with an additional 12 grafted ramets maintained as uninfested controls.

Foliage samples were taken on 16 May 2019 prior to BWA infestation, 14 June 2019, 4 weeks after infestation, and 4 October 2019, 20 weeks after infestation ($n = 312$). Between 6 and 10 cm of branch tissue was collected from each ramet and stored at $-20\ °C$ in a freezer until HIPV extraction. Based on prior observations by Newton et al. [14], 4 weeks roughly corresponds to the period at which mobile BWA crawlers begin feeding on the host and morph into a flat, waxy, neosistentes resting stage [24]. The 20-week sample collection represented, approximately, an entire growing season of BWA feeding on the trees. The clones were grouped into 'Good' (Veitch fir and three Fraser fir clones, $n = 25 \times 3$ sample collections), 'Intermediate' (silver fir and two Fraser fir clones, $n = 18 \times 3$ sample collections), and 'Poor' (seven Fraser fir clones, $n = 49 \times 3$ sample collections) performance groups based on the phenotype assessments reported in Thomas, 2021 [21]. 'Good' performing clones generally had little to no gouting on most ramets with strong apical dominance, good growth rates, and healthy green foliage. 'Poor' performing clones generally had excessive gouting on the majority of ramets with poor growth, loss of apical dominance, and shrubby form.

### 2.2. Extractions and Analytical Methods

Fir needles were excised from collected branches and weighed to approximately 1.5 g. Weighed samples were then placed in 20 mL of pentane solvent in 40 mL glass vials capped with PTFE sealed lids. Capped vials were then wrapped with parafilm (Bemis Company Inc., Neenah, WI, USA) and stored at $-6\ °C$ in a freezer for precisely 28 days. Pentane solutions where then filtered through P5 grade Whatman (Cytiva, Marlborough, MA, USA) filter papers into fresh vials. Finally, exactly 1.5 mL of filtered solution from each sample was pipetted into 2 mL autosampler vial for analysis.

Qualitative analysis of a subset of 14 samples, one per clone, was undertaken on an Agilent 7820A gas chromatograph (GC) equipped with a 5977B mass selective detector (MSD) (Agilent, Santa Clara, CA, USA). Agilent's MassHunter GC/MS Acquisition B.07.00.1413 software and the NIST 2011 library was used for peak identification. Quantitative analysis was undertaken on an Agilent 6890N GC with a flame ionization detector (FID). The column used in both instruments was a 30 m × 0.25 mm (inside diameter) Agilent HP-5 Ultra Inert that possessed a film thickness of 0.25 μm (serial:19091S-433UI HP). The temperature program was modified from Johnson et al. [25] and followed the

protocols established previously by Williams and Avakian [26] and Thomas et al. [27]. This program was optimized for low molecular weight terpenes and used for both instruments. The temperature program for the GC oven was as follows: the initial temperature was held at 80 °C for 2 min and then increased 10 °C/min to the final temperature of 280 °C which was held for an additional two minutes. The injector and detector temperatures were maintained at 250 and 275 °C, respectively. Analytical standards for the terpenes, α-pinene, β-pinene, camphene, caryophyllene, and humulene, were purchased from Sigma-Aldrich (St. Louis, MO, USA). Standards were diluted in pentane to a concentration of 20 µg/mL and then run in triplicate to establish retention times. The retention times of other compounds were based on qualitative GC-MS analysis.

*2.3. Statistical Analysis*

Analysis of foliar HIPVs across the 14 tested clones and the three sample collection periods was carried out via sparse partial least-square regression-discriminant analysis (sPLS-DA) in R statistical software version 4.0.3 [28] with the 'mixOmics' package [29,30]. The number of components was selected based on the minimum balanced error rate (BER) [31]. Accuracy of the sPLS-DA model in predicting an individual ramet's performance group was based on the receiver operating characteristic (ROC) curve [32]. Repeated measures ANOVA and Tukey's honestly significant difference (HSD) posthoc analysis of individual compounds comparing means across performance groups and sampling periods was completed in JMP Pro version 15.2.0 [33].

## 3. Results

Fir foliar extracts contained between 11 and 29 components (see Figure 1 for sample chromatogram). A total of 11 compounds, primarily terpenes and terpenoids, were identified in the fir needle extracts (see Table S1 for complete records). These compounds included α-pinene, camphene, β-pinene, 3-carene, D limonene, maltol, borneol, bornyl acetate, caryophyllene, humulene, and trace amounts of α-bisabolene. The sesquiterpene α-bisabolene was detected in only three samples and was not included in any statistical analysis. Other HIPVs, such as green leaf volatiles (GLVs), may have been present in fir needles but may not have been extracted by the non-polar pentane solvent. GLVs also require a polar column, lower column temperatures, and longer temperature programs for adequate peak separation [34].

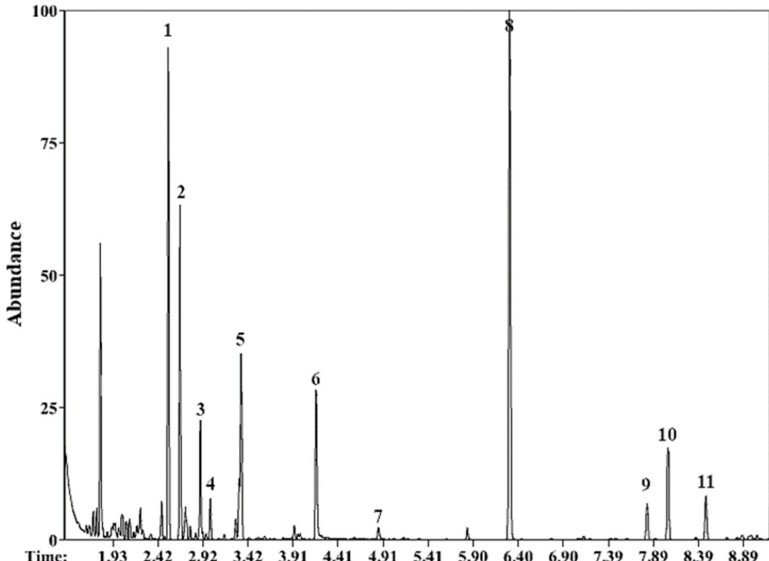

**Figure 1.** Sample chromatogram of fir foliar extract. Peak assignments are as follows: (1) α-pinene, (2) camphene, (3) β-pinene, (4) 3-carene, (5) D limonene, (6) maltol, (7) (-)-borneol, (8) bornyl acetate, (9) caryophyllene, (10) humulene, and (11) α-bisabolene.

sPLS-DA analysis based on six components reveals the sesquiterpenes caryophyllene and humulene are strong predictors of 'Good' performing clones (those with putative resistance to BWA, Figure 2). Reduced foliar concentrations of D limonene and borneol are strong predictors of 'Intermediate' performing clones. Overall, identified HIPVs are only weakly predictive of the clone performance group but are moderately predictive of uninfested control ramets (Figure 3), with a control group AUC of 0.825, a 'Good' performance group AUC of 0.74, an 'Intermediate' performance group AUC of 0.749, and a 'Poor' performance group AUC of 0.727.

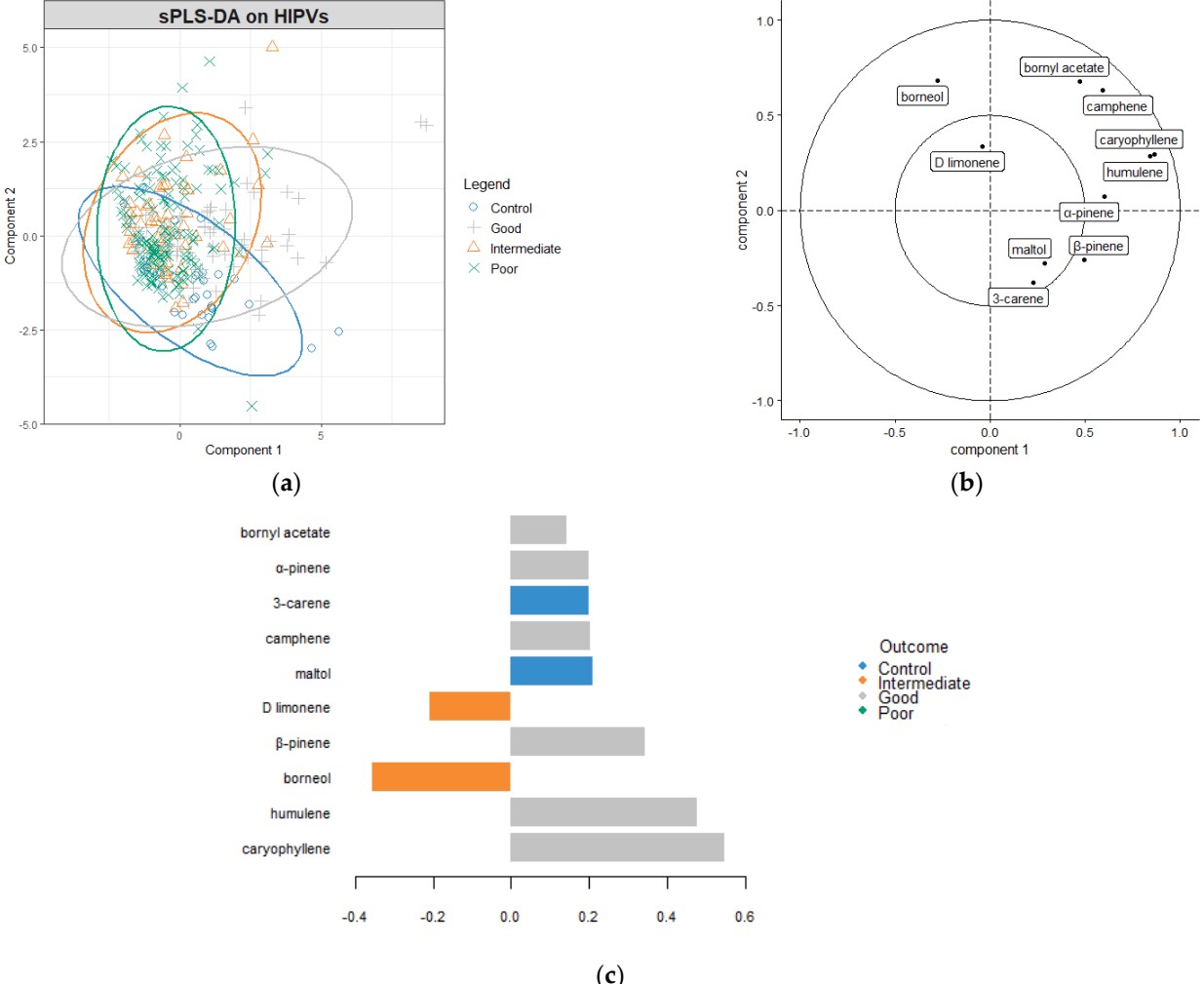

**Figure 2.** Results of sPLS-DA analysis. (**a**) sPLS-DA components 1 and 2, describing 48% of total data variance, (**b**) correlation circle plot on sPLS-DA components 1 and 2, and (**c**) visual representation of sPLS-DA variable contributions based on component 1. Contributions are ranked from the bottom (important) to the top (less important). Color denotes the associated performance group.

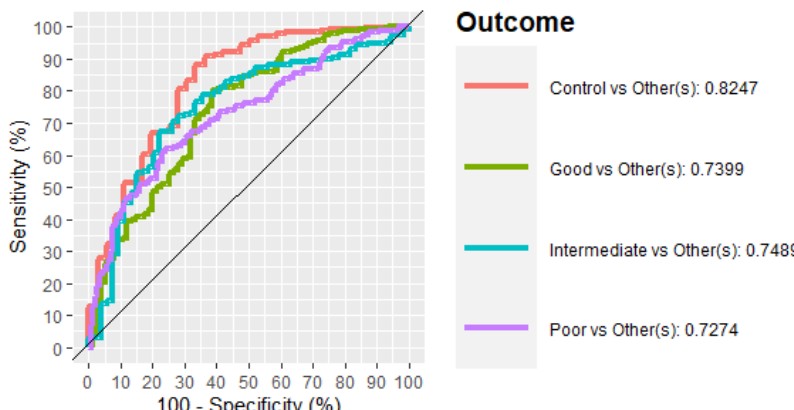

**Figure 3.** Receiver Operating Characteristic (ROC) curve based on six sPLS-DA components predicting sample performance group or control group designation.

Repeated measures ANOVA on individual HIPVs revealed that 9 out of the 10 analyzed compounds had significant differences in foliar concentrations based on performance group, sampling period, or an interaction of the two factors ($\alpha$ = 0.05, Table 1). $\beta$-pinene was the only compound without significant differences among these variables. Three compounds had distinct differences based on Tukey's HSD posthoc analysis. Maltol, the only non-terpenoid HIPV identified, differed significantly only in the pre-infestation control group samples where it was found in significantly greater concentrations than in all infested groups (not shown). The sesquiterpenes caryophyllene and humulene had significantly higher foliar concentrations in the 'Good' performance group at the 20-week sampling period (Figure 4a,b, respectively). The monoterpene camphene and monoterpenoid bornyl acetate generally had increasing foliar concentrations in infested trees over time regardless of performance group, but decreased foliar concentrations over time in uninfested trees (Figure 4c,d, respectively). Slightly increased camphene and bornyl acetate concentrations were noted in 'Good' performing clones but were not distinctly separated from poorer performers by Tukey's HSD posthoc analysis.

**Table 1.** $p$ values for all 10 quantifiable HIPV compounds. Degrees of freedom (df) for each compound are as follows: Performance Group df = 3, Sampling Period df = 2, and Performance Group $\times$ Sampling Period df = 6. Asterisk (*) indicates significant $p$ values ($\alpha$ = 0.05).

| Compound | Performance | | Sampling Period | | Performance $\times$ Sampling Period | |
|---|---|---|---|---|---|---|
| | **F Ratio** | **Prob. > F** | **F Ratio** | **Prob. > F** | **F Ratio** | **Prob. > F** |
| $\alpha$-Pinene | 1.252 | 0.295 | 3.02 | 0.051 | 3.973 | <0.001 * |
| Camphene | 3.267 | 0.025 * | 10.373 | <0.001 * | 4.393 | <0.001 * |
| $\beta$-Pinene | 1.993 | 0.1199 | 1.642 | 0.196 | 1.194 | 0.311 |
| 3-Carene | 4.507 | 0.005 * | 7.337 | <0.001 * | 2.829 | 0.012 * |
| D limonene | 2.377 | 0.074 | 6.862 | 0.001 * | 2.551 | 0.021 * |
| Maltol | 9.734 | <0.001 * | 8.3 | <0.001 * | 4.352 | <0.001 * |
| Borneol | 6.448 | <0.001 * | 5.605 | 0.004 * | 3.381 | 0.003 * |
| Bornyl acetate | 1.715 | 0.169 | 3.414 | 0.035 * | 2.434 | 0.027 * |
| Caryophyllene | 9.459 | <0.001 * | 10.874 | <0.001 * | 6.675 | <0.001 * |
| Humulene | 10.184 | <0.001 * | 10.109 | <0.001 * | 8.247 | <0.001 * |

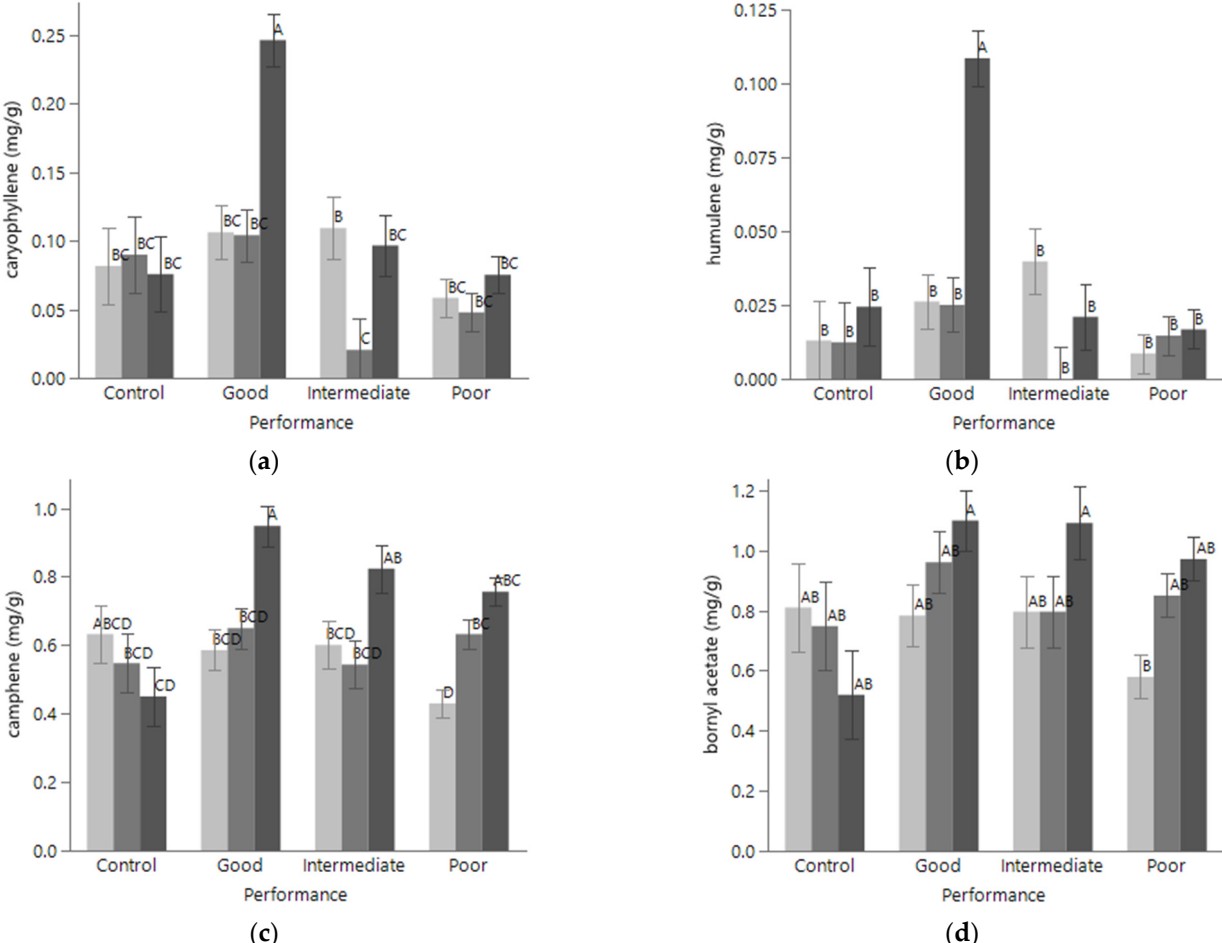

**Figure 4.** Significant differences in the concentrations of select terpenoids across Balsam woolly adelgid (BWA) infested groups over three sampling periods: light grey = pre-infestation, medium grey = 4 weeks after infestation, dark grey = 20 weeks after infestation. (**a**) Caryophyllene mg/g needle tissue, (**b**) humulene mg/g needle tissue, (**c**) camphene mg/g needle tissue, and (**d**) bornyl acetate mg/g needle tissue. Error bars show standard errors. Means with different letters are significantly different (Tukey's HSD, $\alpha \leq 0.05$).

## 4. Discussion

Foliar HIPVs were weakly to moderately predictive of Fraser fir clone performance under BWA attack. Most importantly, the sesquiterpenes caryophyllene and humulene were the primary correlates with the 'Good' performing clones and were strongly induced only in 'Good' clones after 20 weeks of BWA infestation. This correlation between high sesquiterpene concentrations and good clone performance when infested with BWA is in line with previous findings in wild Fraser fir populations by Carlow et al. [18].

Two compounds, bornyl acetate and camphene, were induced over time in all infested Fraser firs as well as silver fir and Veitch fir in response to BWA infestation. These compounds were not induced in any uninfested control trees. This finding indicates that the induction of bornyl acetate and camphene may be part of a general response to BWA attack, although the correlation between the induction of these compounds and the performance of the host Fraser fir is less clear in this study. The induction of camphene is significant after 20 weeks of BWA infestation in 'Good' performing Fraser fir clones, but not clearly distinct from the camphene response observed in 'Intermediate' and 'Poor' performing clones in this study. However, previous work by Grégoire et al. [19] found that increased foliar camphene concentrations were significantly correlated with lower gouting response

in the closely related balsam fir, suggesting that increased camphene concentrations may still be important to host fir performance.

Not all findings in this study matched those of previous terpenoid analyses in BWA-infested firs. In contrast to previous findings by Carlow et al. [18], no clear correlation of 3-carene or β-pinene with clone performance was found. In fact, β-pinene did not differ significantly between performance groups or controls and did not change over the course of the growing season in this study. It is not entirely clear why the response of these monoterpenes differed between the studies. One possibility is that HIPVs can be induced by abiotic environmental stressors [3] that are much more prevalent in natural stands of Fraser fir. Additionally, no juvabione was detected in fir foliage in this study despite previous research finding significant correlations between induced juvabione and BWA infestation [20]. Methods used in this study were not optimized for the detection and quantification of juvabione, which occurs primarily in Fraser fir bark. However, α-bisaboline was detected in trace amounts in several clones as well as Veitch fir and is believed to be the precursor to juvabione and related juvenile hormone mimics in all *Abies* species [35]. Additional research targeting the terpenoid juvabione and other juvenile hormone mimics is needed to determine their contributions to the putative BWA resistance of the Fraser fir clones used in this study.

The terpenes and terpenoids found in this study fill diverse roles in plant defense against insects across the plant kingdom. An indirect defense mechanism related to induced terpenoids in 'Good' performing firs is unlikely as no major predators or parasitoids of BWA are known from the Southern Appalachians. Terpenes and terpenoids imparting resistance are most likely repellent or toxic to BWA. Bornyl acetate was the focus of previous work by Bucholz et al. [36] as the compound has known insecticidal properties. While bornyl acetate effectively controls populations of some aphid species, it does not significantly impact BWA egg eclosion, survivorship, or fecundity on Fraser fir and is therefore unlikely to play a role in direct defense against the adelgid. Caryophyllene and humulene are also known to have toxic effects on some insect herbivores [37] and the potential toxic or deterrent effects of these compounds have not yet been evaluated on BWA. Based on adelgid numbers observed in this study and reported previously by Thomas, 2021 [21], adult adelgid numbers were similar between 'Good' and 'Poor' performing clones at the end of the growing season. However, BWA egg eclosion could still be affected by the foliar concentrations of these sesquiterpenoids.

## 5. Conclusions

Several terpene and terpenoid compounds are induced in the fir host by BWA attack as ascertained by gas chromatographic and statistical analyses. The compounds camphene and bornyl acetate appear to be induced in all Fraser firs, silver firs, and Veitch firs infested with BWA regardless of tree performance. The sesquiterpenes caryophyllene and humulene are induced only in high performing Fraser firs and Veitch fir after sustained attack by BWA over the course of a growing season. Together, these four compounds appear to be part of a significant defensive response to BWA by the host fir tree. Ultimately, it remains unclear if this differential HIPV response observed in 'Good' performing fir clones is a primary mode of putative resistance in the firs or if this phytochemical response simply correlates with an effective physiological response by the host plant to BWA through other means. Further research on the direct effects of these terpenoids on BWA is needed to establish causality. However, screening for the induction of these terpenoid compounds in infested Fraser firs could be a useful tool for selecting resistant individuals for use in future resistance breeding programs.

**Supplementary Materials:** The following supporting information can be downloaded at: https://www.mdpi.com/article/10.3390/f13050716/s1, Table S1.

**Author Contributions:** Conceptualization, A.T., R.M.J., J.F.; methodology, A.T., D.C.T.; formal analysis, A.T.; investigation, A.T.; resources, A.T., D.C.T. and R.M.J.; data curation, A.T.; writing—original draft preparation, A.T.; writing—review and editing, A.T., D.C.T. and R.M.J.; visualization, A.T.; supervision, D.C.T. and R.M.J., project administration, R.M.J. All authors have read and agreed to the published version of the manuscript.

**Funding:** This research has been funded by the North Carolina Specialty Crops Block Grant Program (Grant number 19-019-4019), and a 2019 grant from the North Carolina Christmas Tree Growers Association.

**Data Availability Statement:** Not applicable.

**Acknowledgments:** The authors would like to thank Ben Smith, Joey Borders, and Andy Whittier for assistance with sample collection and Ambre Chiomento for assistance with sample analysis.

**Conflicts of Interest:** The authors declare no conflict of interest.

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
