# Peer review of "Sesquiterpene Induction by the Balsam Woolly Adelgid (Adelges piceae) in Putatively Resistant Fraser Fir (Abies fraseri)"

_forests, doi:10.3390/f13050716_

Round 1
Reviewer 1 Report
Forests manuscript - -
General comments:
In general, the manuscript titled Sesquiterpene induction by the Balsam woolly adelgid (adelges picea) in putatively resistant Fraser Fir (Abies fraseri) has a valuable topic. The manuscript is well written. The English language and style are fine except for moderate English language check required. The experimental design is adequate. My main concern is the discussion section.
There are some minor comments.
Detailed comments:
In general, please avoid using personal pronouns such as line 230 (our study) and apply this rule throughout the manuscript
Keywords:
Please add Resistance to the keywords list.
Abstract:
The aim of the study and the main objectives were not clearly stated.
Please state the aim of this study clearly in this section.
Introduction:
This section didn’t provide enough background about the topic. The introduction needs to be elongated and enriched.
Materials and Methods:
The experimental design is adequate and suitable to the current study BUT too many details were stated. Please provide only the necessary details followed by the corresponding citation.
Results:
The results were well presented BUT with a poor discussion.
Discussion:
This section is poorly written
As mentioned before, this section is poorly written. I had a hard time to relate the discussion section with the corresponding data in the tables and the figures. There are 4 figures and one table that hardly mentioned and poorly discussed.
*Please rewrite this section and provide the appropriate citations in argument, and valuable discussion to the current results. If it will help, the author is advised to combine the results section and the discussion section.
Conclusion:
This section is ok. This section provides a good conclusion for the study and includes the significant findings with some recommendations for further study about this point.
References:
The authors provided enough citations, and it was UpToDate.
Author Response
Detailed comments:
Personal pronouns were removed throughout the document.
Keywords:
'resistance' was added to the keywords.
Abstract:
Objective has been clearly stated in the Abstract (see lines 13 and 14 in the revised manuscript).
Introduction:
Added additional background to the introduction on the following subjects: BWA effects on host fir (line 60), implications for host fir genetics (line 68), and the relationship of genetic conservation to the Christmas tree industry (line 96).
Materials and Methods:
Redundant information has been removed. Extraneous information about RNASeq study was removed (this has also been removed from the discussion section).
A few clarifications about clone selections have been added based on Review 2's suggestions (line 100).
Discussion:
While we opted not to combine the results and discussion section, we acknowledge there were significant errors and a lack of clarity in the discussion section. We have extensively rewritten the discussion section to clarify our findings but did not change the overall interpretation of our findings.
Reviewer 2 Report
Please, see attach document.

Author Response
All minor grammatical errors, capitalization, and punctuation corrections have been addressed as suggested by the reviewer.
Regarding abbreviations for taxonomic authority: where established abbreviations for authorities exist, they are used. Several taxonomic authorities do not have established abbreviations.
The discussion section has been extensively re-written based on reviewer one's comments and should also better address review two's concerns now.
In response to Reviewer two's comments on the F1 generation: See line 102 in the methods section for clarification on Fraser fir selections. Improved Fraser fir selections were not bred for or selected for BWA resistance prior to this study.